# Titanium-Enriched Medium Promotes Environment-Induced Epigenetic Machinery Changes in Human Endothelial Cells

**DOI:** 10.3390/jfb14030131

**Published:** 2023-02-27

**Authors:** Célio Júnior da C. Fernandes, Rodrigo A. Foganholi da Silva, Patrícia F. Wood, Marcel Rodrigues Ferreira, Gerson S. de Almeida, Julia Ferreira de Moraes, Fábio J. Bezerra, Willian F. Zambuzzi

**Affiliations:** 1Laboratory of Bioassays and Cellular Dynamics, Department of Chemical and Biological Sciences, Institute of Biosciences, UNESP—São Paulo State University, Botucatu 18618-970, SP, Brazil; 2Department of Dentistry, University of Taubaté, Taubaté 12020-340, SP, Brazil; 3Program in Environmental and Experimental Pathology, Paulista University, São Paulo 04026-002, SP, Brazil

**Keywords:** biomaterial, titanium, biological analysis, angiogenesis, endothelial cell, epigenetic

## Abstract

It is important to understand whether endothelial cells are epigenetically affected by titanium-enriched media when angiogenesis is required during bone development and it is expected to be recapitulated during osseointegration of biomaterials. To better address this issue, titanium-enriched medium was obtained from incubation of titanium discs for up to 24 h as recommended by ISO 10993-5:2016, and further used to expose human umbilical vein endothelial cells (HUVECs) for up to 72 h, when the samples were properly harvested to allow molecular analysis and epigenetics. In general, our data show an important repertoire of epigenetic players in endothelial cells responding to titanium, reinforcing protein related to the metabolism of acetyl and methyl groups, as follows: Histone deacetylases (HDACs) and NAD-dependent deacetylase sirtuin-1 (Sirt1), DNA methyltransferases (DNMTs) and ten-eleven translocation (TET) methylcytosine dioxygenases, which in conjunction culminate in driving chromatin condensation and the methylation profile of DNA strands, respectively. Taking our data into consideration, HDAC6 emerges as important player of this environment-induced epigenetic mechanism in endothelial cells, while Sirt1 is required in response to stimulation of reactive oxygen species (ROS) production, as its modulation is relevant to vasculature surrounding implanted devices. Collectively, all these findings support the hypothesis that titanium keeps the surrounding microenvironment dynamically active and so affects the performance of endothelial cells by modulating epigenetics. Specifically, this study shows the relevance of HDAC6 as a player in this process, possibly correlated with the cytoskeleton rearrangement of those cells. Furthermore, as those enzymes are druggable, it opens new perspectives to consider the use of small molecules to modulate their activities as a biotechnological tool in order to improve angiogenesis and accelerate bone growth with benefits of a fast recovery time for patients.

## 1. Introduction

The application of biomaterials has been widely proposed within the fields of tissue engineering and regenerative medicine, as well as to recover the functional performance in edentulism [1,2]. However, a significant number of patients may experience biological and mechanical complications characterized by surrounding tissue degradation and consequently the failure of implants [3]. In an attempt to minimize the failure of implants, numerous researchers have endeavored to elucidate the physiological and molecular mechanisms of osteointegration [4,5,6,7,8]. In this sense, biological data have guided the proposal of new surfaces presenting biomimetic properties predicted to develop necessary biological responses and so impacting the performance of surrounding host tissue during osteointegration [9,10].

Among the most commonly used biomaterials within the biomedical area, titanium has been listed as the most usual metallic alloy in biomedical applications. Indeed, titanium has well-accepted properties in biocompatibility scenarios which have repercussions on clinical success [11]. The biological performance of materials used in medicine and dentistry requires biological responses upon signaling related to the interaction of cells and the physicochemical properties of their surfaces [12,13,14,15,16,17].

Among the players in regenerative tissue surrounding implanted devices, there is a pleiotropic lineage of cells, which work together to promote new bone formation [18]. In this scenario, it is relevant to consider the coupling between endothelial cells and osteoblasts, which is known to compromise osteogenesis [19,20]. Although knowledge has been improved considering osteoblasts responding to materials [21], the relevance of endothelial cells responding within this scenario is barely known [22,23,24,25,26]. During the biomaterial-related osseointegration mechanism, it is expected that endothelial cells recapitulate the coupling with undifferentiated cells in order to optimize the osteogenesis and culminate in the success of osseointegration. Although blood vessels are not in close contact with implanted material surfaces, we have recently reported that titanium-based devices release a considerable number of molecules which interfere with cell metabolism. Importantly, the epigenetic mechanism drives the capacity of cells to respond to microenvironment changes. Epigenetics is the study of changes in gene expression without alterations of DNA itself, which include the reactions to modulate the incorporation of methyl groups into DNA strands as well as acetyl, sumoyl, and phospho groups into histones [27]. Additionally, epigenetic regulation of histone acetylation and DNA methylation are pivotal points involved in cellular responses during cell adhesion and viability [28]. Conversely, histone deacetylases (HDACs) affect gene expression in cells grown on implant surfaces [29]. Recently, differences in global DNA methylation profiles were also demonstrated with relevance to genetic expression of osteogenic pathways in mesenchymal stem cells responding to titanium [30]. Generally, this mechanism has been shown to have high performance with wound healing potential, with properties ranging from high predictability to later cellular mechanisms involved in osseointegration. We reported earlier the biological responses of cells reacting to titanium and nanohydroxyapatite-blasted surfaces [31,32,33]. In this regard, tissue integration depends on biomaterial microstructure and composition, mainly considering the properties of the surfaces of materials, and the effect of the pore size of titanium on the viability and function of fibroblasts and tenocytes in a dynamic bioreactor was evaluated [34].

Again, the knowledge of how endothelial cells react to titanium alloys remains elusive although progress has been made considering osteoblasts [15,35,36,37]. Endothelial cells are involved in angiogenesis, which is essential to deliver nutrients and oxygen during the healing process around biomaterial, as well as contribute to osteogenesis by coupling the paracrine mechanism between themselves and osteoblasts. To the best of our knowledge, this is the first study intending to explore the epigenetic machinery in endothelial cells exposed to titanium-enriched media. Altogether, it involves epigenetic changes in human endothelial cells by processing genes and enzymes and these data are gathered to draw a complimentary molecular mechanism in wound healing affected by titanium-based devices in order to better predict the mechanisms involved in osteointegration of biomaterials into bone.

## 2. Materials and Methods

### 2.1. Reagents

The titanium discs were kindly donated by S.I.N Implants System Company (SIN) (São Paulo, Brazil). Antibodies: HDAC1 (10E2) Mouse mAb #5356; HDAC2 (3F3) Mouse mAb #5113; HDAC3 (7G6C5) Mouse mAb #3949; HDAC6 (D2E5) Rabbit mAb #7558; SirT1 (1F3) Mouse mAb #8469; GAPDH (D16H11) XP^®^ Rabbit mAb #5174, Life Technologies/Molecular Probes, Inc. (Eugene, OR, USA). DNMT1 Antibody (60B1220.1), DNMT3A Antibody (64B1446) (Novus Biologicals, LLC, Centennial, CO, USA), TET3 Antibody, Anti-DNMT3B (orb372330), TET1 (orb228563), and TET2 (orb131790) were purchased from BiorByt (San Francisco, CA, USA).

### 2.2. Titanium-Enriched Medium and Experimental Design

Titanium discs were previously prepared to subject their surfaces to dual acid-etching (DAE; w_Ti) and they were compared to machined surfaces (wo_Ti) [38]. Technically, to obtain the titanium-enriched medium from the samples, discs (0.6 cm diameter) were incubated within conic tubes containing cell culture medium (0.2 g/mL *w*/*v*, without fetal bovine serum (FBS)) for up to 24 h as recommended by ISO 10993-5:2016 (Figure 1a). Thereafter, this titanium-enriched medium was used to challenge human endothelial cells for up to 72 h, when the cells were harvested for biological analysis of the epigenetic mechanism (Figure 1b). Figure 1b depicts DNA methylation and histone acetylation molecular mechanisms. Additionally, both gene activation and repression are regulated by the pattern of acetylation of core histones, and the histone acetylation is mediated by coactivators (histone acetyltransferase: HATs) and corepressors (histone deacetylases: HDACs).

### 2.3. Cell Culture

To evaluate whether there are changes in endothelial cells responding to Ti-enriched medium and further correlation with aspects of angiogenesis, human umbilical vein endothelial cells (HUVECs; ATCC) were used in this study. Throughout the experiments, cells were grown in DMEM containing antibiotics (100 U/mL penicillin, 100 mg/mL streptomycin), and supplemented with 10% fetal bovine serum (Nutricell, Campinas, SP, Brazil). Cells were maintained at 37 °C, 5% CO_2,_ and 95% humidity. Importantly, these cells were mycoplasma-free and were used in this study in accordance with the manufacturer’s recommendations.

### 2.4. Gene Expression

#### 2.4.1. RNA Isolation and cDNA Synthesis

The total RNA was extracted using Ambion TRIzol Reagent (Life Sciences—Fisher Scientific Inc, Waltham, MA, USA) and its concentration and purity determined by measuring the absorbance at 260/280 nm (NANODROP 2000, Thermo, Waltham, MA, USA). After treatment of total RNA with DNase I (Invitrogen, Carlsbad, CA, USA), the cDNA was synthesized from 2 μg total RNA with a High-Capacity cDNA Reverse Transcription Kit (Applied Biosystems, Foster City, CA, USA) in a standard 20 μL volume according to the protocol.

#### 2.4.2. Gene Expression

Gene expression in endothelial cells was measured using SYBR Green quantitative PCR analysis. Real-time PCR amplification was performed with the QuantStudio^®^ 3 Real-Time PCR system (Thermo Fisher Scientific, Waltham, MA, USA). The 10 μL PCR system consisted of 1 μL cDNA (50 ng), 0.5 μmol forward primer, 0.5 μmol reverse primer, 5 μL SYBR Green qPCR mix, and 3 μL ultrapure DEPC-treated water. PCR primers were designed using Primer3 Input (version 0.4.0) software. The secondary structure and annealing temperature were analyzed by the Beacon Designer program, Free Edition (http://www.premierbiosoft.com/, accessed on 29 July 2021) and confirmed by in silico PCR (https://genome.ucsc.edu/, accessed on 29 July 2021). Primer 5.0 software and primer synthesis were acquired from Invitrogen Co. (Life Technologies, Carlsbad, Califórnia, EUA). Primer sequences and CR conditions are presented in Table 1. Gene expressions are expressed as compared to control cells by the ^ΔΔ^CT method, using Gapdh as a housekeeping control in three independent experiments in duplicate. Finally, cDNA was used to evaluate each gene’s expression (Table 1).

### 2.5. Western Blot

Cultures were exposed to titanium-enriched medium for up to 72 h, when the cells were lysed using lysis buffer (50 mM Tris-HCl, pH 7.4, 1% Tween 20, 0.25% sodium deoxycholate, 150 mM NaCl, 1 mM EGTA, 1 mM *O*-vanadate, 1 mM NaF, and protease inhibitors (1 μg/mL aprotinin, 10 μg/mL leupeptin, and 1 mM aminoethylfluorosilicon 4-fluoride hydrochloride)) for up to 2 h in ice and thereafter sonicated, then the samples were centrifuged to collect the supernatant and the pellet was discarded. The protein concentration was determined by the Lowry method and an equal volume of Laemmli buffer (2X sodium dodecyl sulfate (SDS), 100 mM Tris-HCl (pH 6.8), 200 mM dithiothreitol (DTT), 4% SDS, 0.1% bromophenol blue, and 20% glycerol) was added. The obtained pool of proteins was boiled for 5 min at 95 °C in denaturant conditions. To resolve the proteins into the gel, the proteins were resolved into SDS–PAGE (8% or 10%), then were transferred to PVDF membranes (Millipore), with sequential incubation with blocking solution containing 1% bovine serum albumin in Tris-buffered saline (TBST, Tween 20 (0.05%)) for 1 h. Thereafter, the blocked membranes were subjected to specific primary antibody incubation for up to 12 h, at 4 °C. Afterward, the membranes were extensively washed using TBS–Tween 20 (0.05%; 3X), then they were incubated with secondary antibodies for 1 h at room temperature, and washed using TBS. Enhanced chemiluminescence solution (ECL) was finally used to detect the bands. GAPDH was used as a housekeeping and loading control.

### 2.6. Statistical Analysis

Bands of immunoblots were measured using densitometry tools and analysis to obtain arbitrary values, which represent mean ± standard deviation (SD) in the graphs. Statistical analyses were applied using Student’s *t*-test (2-tailed) with *p* < 0.05 considered statistically significant and *p* < 0.001 considered highly significant. Accordingly, we used one-way ANOVA with a Bonferroni post-test in order to compare all pairs of groups. In this case, the significance level was α = 0.05 (95% confidence interval). The software used was GraphPad Prism 6.

## 3. Results

Epigenetic mechanisms create a bridge between the microenvironmental properties and capacity to activate genes. Specifically, we have focused on investigating the epigenetic-related mechanisms in endothelial cells responding to Ti-enriched medium by evaluating the profile of expression of enzymes related to (de)methylation of DNA and histone (de)acetylation. Thus, Figure 2 shows the panorama of histone acetylation processes by exemplifying genes encoding the main HDAC enzymes involved in the epigenetic mechanisms. The family of genes encoding HDACs in endothelial cells responding to Ti-enriched medium was evaluated, and our data show a significant reduction in gene expression of HDAC1 (Figure 2a,d), HDAC2 (Figure 2b,e), and HDAC3 (Figure 2c,f) when compared to the control group. Importantly, the protein synthesis from those encoding genes presents a very similar profile to their respective transcript profiles. Conversely, there is a significantly higher expression of the HDAC6 gene in endothelial cells responding to w_Ti (Figure 3a), and this was corroborated with protein evaluation performed by Western blot (Figure 3c), which opens new perspectives to consider HDAC6′s relevance in this scenario. It is important to mention that SIRT1 was kept lower in the cultures responding to both wo_Ti and w_Ti considering its transcript profile (Figure 3b), however, the protein seems to cause a constitutive effect in endothelial cells (Figure 3d). Importantly, a significantly higher expression of H2B was observed in response to both wo_Ti and w_Ti (Figure 4).

Thereafter, we addressed the involvement in epigenetic mechanisms displayed by DNA methyltransferases (DNMTs) and TETs, both related to regulating DNA methylation (Figure 5). Our data show that the expression of the DNMT1 gene was around 6-fold higher in response to w_Ti than in the control cultures (Figure 5a), while both the DNMT3A and DNMT3B genes were downregulated (Figure 5b,c). However, this transcriptional profile does not reflect on the protein synthesis as it was lower in endothelial cells responding to w_Ti (Figure 5d,g), while DNMT3A (Figure 5e,h) and 3B (Figure 5f,i) protein levels seem to reflect the performance of those genes.

We also investigated both TET1 and TET2 genes and related proteins by using qPCR and Western blot technologies, respectively. TET1 and TET3 genes remain unchanged (Figure 6a,c) while the TET2 gene was more highly expressed (~2.5-fold change) than in the control cultures of endothelial cells responding to both wo_Ti and w_Ti (Figure 6b), although protein content shows a significant decrease in both TET1 and 2 (Figure 6d,e).

## 4. Discussion

During the osteointegration process of bone growth along metallic devices, well-controlled angiogenesis is expected by stimulating endothelial cell viability and proliferation, supplying the newly formed bone with cells, nutrients, and metabolite changes. Previously, we have shown that titanium-based implants induce dynamic microenvironment changes by releasing titanium molecules [39], which are very active in bone cells [40]. Considering the heterogenicity and plasticity of cells surrounding implants, it is necessary to understand the behavior of endothelial cells in this complex biological scenario related to wound healing. As seminal works have shown epigenetic marks governing the capacity of cells to respond to environmental changes [41], we have addressed this issue in this study by investigating the requirement of enzymes able to control the (de)methylation of DNA strands as well as (de)acetylation of histones, which directly compromises the capacity of cells in activating specific genes in response to the environment changes, as well as impacts their capacity to respond to biomaterial effects.

Endothelial cells compose not only a barrier in blood vessels but also play multiple roles in various pathophysiological events, such as the control of vasculature tonus and capacity to interact with the blood and surrounding cells at the same time, contributing to major signaling able to drive angiogenesis [42]. Angiogenesis is an essential event expected during biomaterial interaction with the host tissues and cells and determines the success or failure of dental implants during biological steps of osseointegration. We have, over recent years, stated that titanium-based dental implants modulate cells’ specific signaling pathways, most of the time requiring the control of reactive oxygen species (ROS) [43]. Additionally, it is known that ROS drive the activity of relevant proteins intracellularly such as protein tyrosine phosphatases (PTPs), of which PTP1B has distinguished control in osteoblast cell adhesion [43]. Although some progress has been made in this regard, the capacity of titanium-based implants in modifying the microenvironment is barely known and it brings new perspectives to know the role of epigenetic machinery in cells responding to it. Among others, our data show a potential constitutive expression of SIRT1 in endothelial cells, and it might be required to modulate ROS, as ROS production was previously reported in response to a titanium-enriched medium and oxidative stress to compromise the vascular function of SIRT1 [44].

Additionally, the capacity of endothelial cells to modulate gene expression was also investigated here by assaying HDACs, DNMTs, and TETs. Firstly, considering the involvement of histone lysine acetylation balance, our data show a likely pivotal role of HDAC6 in endothelial cells responding to titanium and it is shown to spread through the cytoplasm [45,46]. It is time to correlate HDAC6 with cytoskeleton rearrangement and consider it during cell adhesion, migration, and proliferation. In fact, HDAC6 has been previously reported as contributing to cytoskeleton rearrangement of endothelial cells with important repercussions on angiogenesis [47]. Previously, we have extensively shown that a titanium-enriched medium affects the molecular machinery related to cytoskeleton rearrangement upon integrin activation and, in conjunction, these findings reveal a critical role for HDAC6 in mediating titanium-induced cellular events.

DNMT1, DNMT3A, and TET2 genes were more highly expressed in endothelial cells responding to titanium-enriched medium, but this does not necessarily reflect the protein pattern. Mechanistically, this discrepancy between gene activity and protein profile strongly suggests an efficient post-transcriptional machinery in endothelial cells such as microRNAs and long non-coding RNA (lnc-RNA). Specifically, DNMT1 is a conserved methyltransferase necessary to maintain global methylation of DNA strands, while DNMT3A and DNMT3B are methyltransferases produced de novo [41,48,49]. On the other hand, TETs are involved in an active DNA demethylation pathway by converting 5-methylcytosine (5mC) to 5-hydroxymethylcytosine (5hmC) [50,51,52] and they are important to demethylate DNA of cells responding to environmental changes, such as titanium or other metallic biomaterials used in medicine or dentistry. Additionally, this might be explored in predictable tools such as bioreactors responding to implantable biomaterials [53,54,55,56] as cultures of blood vessels in bioreactors have been reported [57,58]. Additionally, to know the effects of titanium in cells and humans within these fields is also relevant in considering novel propositions in bioengineering research and, as these enzymes are druggable, they are targets for specific design of small molecules able to modulate their activities, aiming at enhancing angiogenesis, and they might accelerate the recovery of patients by improving the osteogenesis surrounding titanium-based implants.

## 5. Conclusions

These findings support the hypothesis that titanium promotes surrounding microenvironment changes and it affects the capacity of endothelial cells to respond to the microenvironment by modulating epigenetic players, among which HDAC6 seems to have a crucial role. In this perspective, whether HDAC6 modulates endothelial cell cytoskeleton rearrangement during angiogenesis during titanium osseointegration can be evaluated.

## Figures and Tables

**Figure 1 jfb-14-00131-f001:**
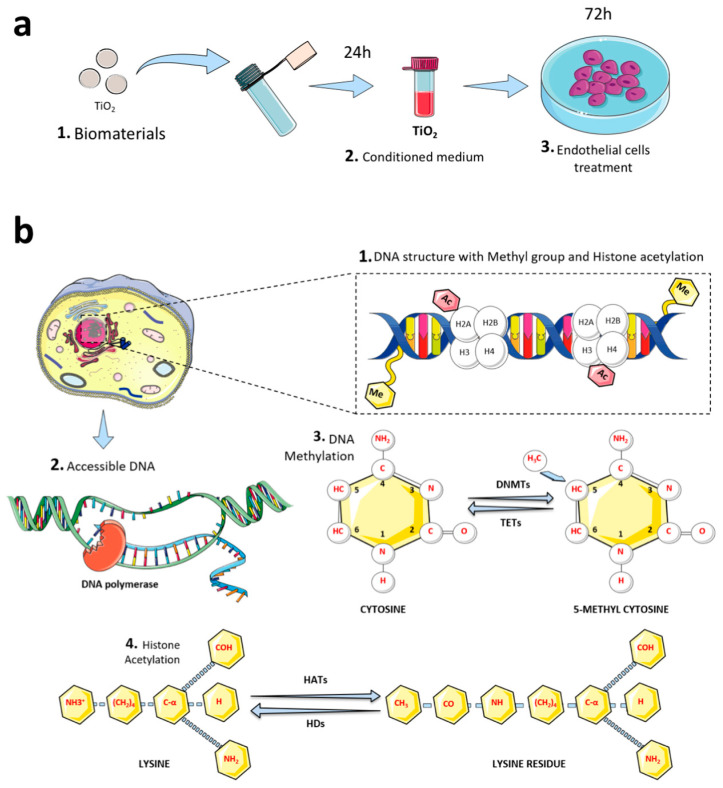
Experimental design performed in this study. (**a**) In order to analyze whether Ti-enriched medium is able to modulate the epigenetic marks in endothelial cells, discs of titanium alloys were incubated in cell culture medium without FBS for up to 24 h, as recommended by ISO 109993-5:2016. Thereafter, the conditioned medium (titanium-enriched medium) was used further to treat endothelial cells for up to 72 h, when the cells were harvested to allow molecular analysis. The experiment was performed in triplicate (n = 3); (**b**) the epigenetic machinery evaluated in this study: Chromatin condensation by evaluating both gene expression and protein level of HDACs (4), and the metabolism of methyl moiety by evaluating DNMTs and TETs (1–3). The discs of titanium evaluated in this study were subjected to DAE (w_Ti) or not (wo_Ti).

**Figure 2 jfb-14-00131-f002:**
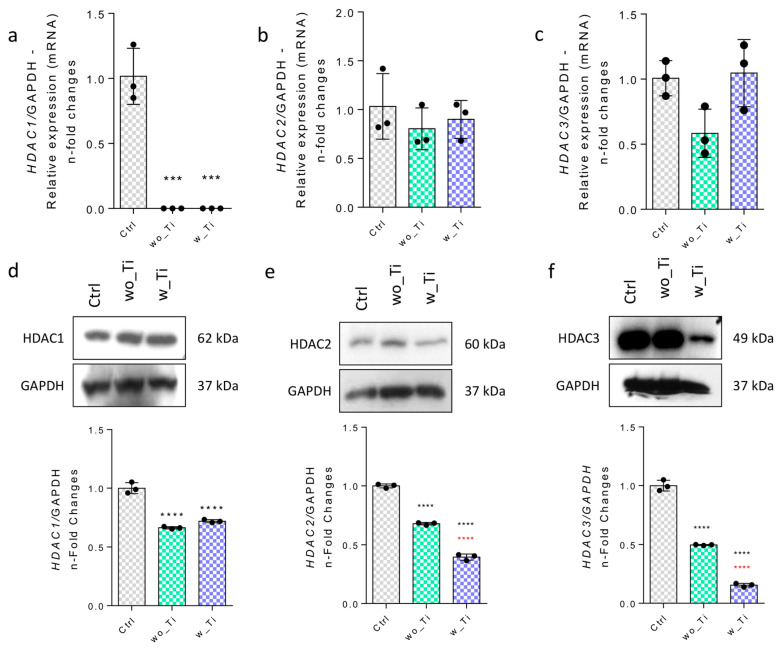
Effect of titanium-enriched medium on HDAC1, HDAC2, and HDAC3. The effect of Ti-enriched medium on the expression of genes related to histone acetylation. HDAC1 (**a**), HDAC2 (**b**), HDAC3 (**c**) were evaluated by qPCR. The control group (Ctrl) values were normalized to 1 and the relative values obtained for wo_Ti or w_Ti are shown as fold changes. The effect was later evaluated regarding the protein level, and the profile was very similar to the transcripts, as follows: HDAC1 (**d**), HDAC2 (**e**), and HDAC3 (**f**). Representative blots as well as the analysis of the densitometry of the bands are shown. Note: HDACs: histone deacetylases; Ctrl: control group; wo_Ti: medium enriched with titanium without DAE; w_Ti: medium enriched with titanium with DAE. *** *p* < 0.0002, **** *p* < 0.0001 when compared with the control (Ctrl) group; and **** *p* < 0.0001 when comparing wo_Ti with w_Ti. The experiment was performed in triplicate (n = 3).

**Figure 3 jfb-14-00131-f003:**
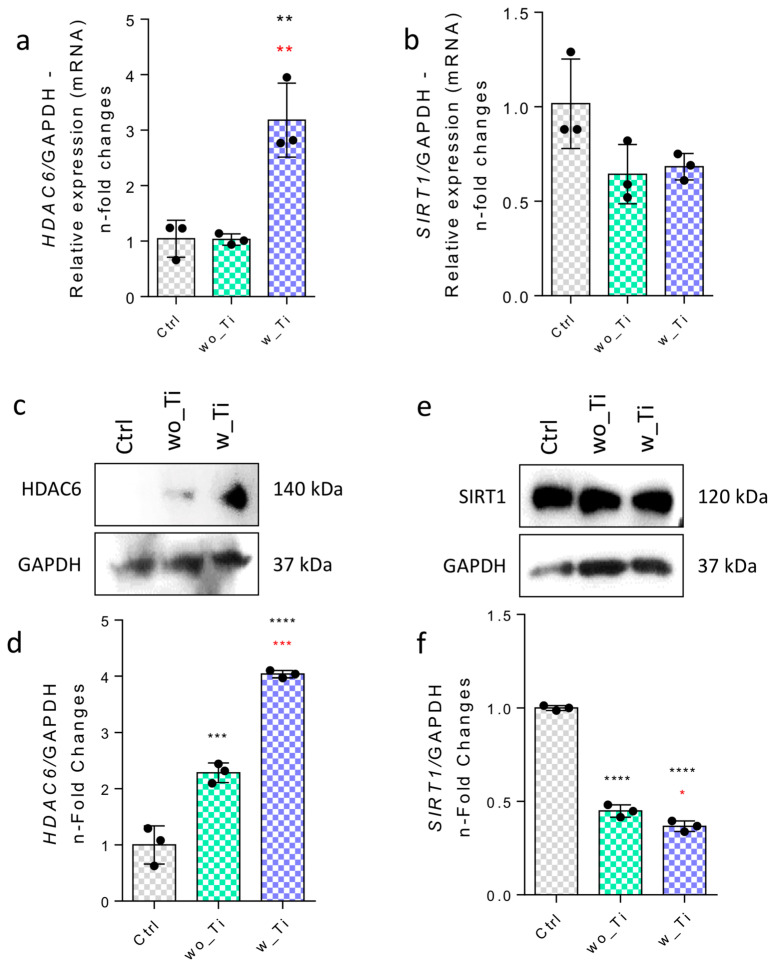
Effect of titanium-enriched medium on HDAC6 and SIRT1. The effect of Ti-enriched medium on the expression of HDAC6 (**a**) and SIRT1 (**b**) was evaluated by qPCR, and the protein expression by Western blot was measured (**c**,**d**,**e**,**f**, respectively). Differences were considered statistically significant when ** *p* < 0.001, *** *p* < 0.0002, **** *p* < 0.0001 when compared with the control (Ctrl) group; and * *p* < 0.04, ** *p* < 0.001, *** *p* < 0.0002 when comparing wo_Ti with w_Ti. Note: HDACs: histone deacetylases, SIRT1: Sirtuin 1, Ctrl: control group. GADPH (gene and protein) was considered a housekeeping gene in this study. The experiment was performed in triplicate (n = 3).

**Figure 4 jfb-14-00131-f004:**
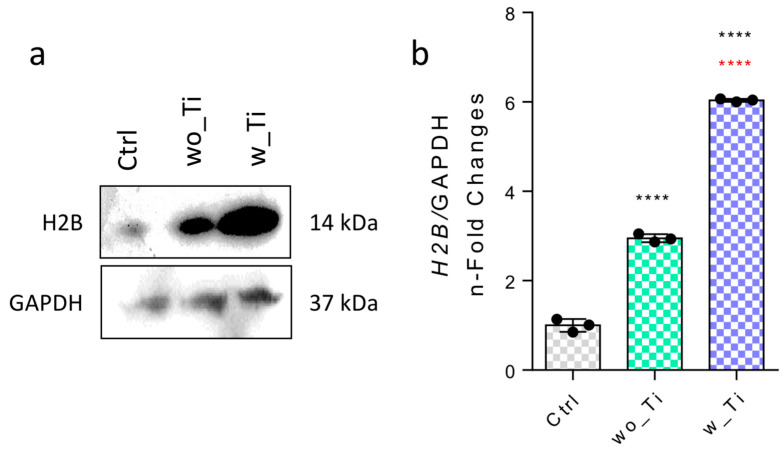
Effect of titanium-enriched medium on H2B. The effect of Ti-enriched medium on the expression of H2B was evaluated by performing Western blot (**a**), densitometry and statistics are also shown (**b**). GAPDH was a loading control. **** *p* < 0.0001 when compared with the control (Ctrl) group; and **** *p* < 0.0001 when comparing wo_Ti with w_Ti. The experiment was performed in triplicate (n = 3).

**Figure 5 jfb-14-00131-f005:**
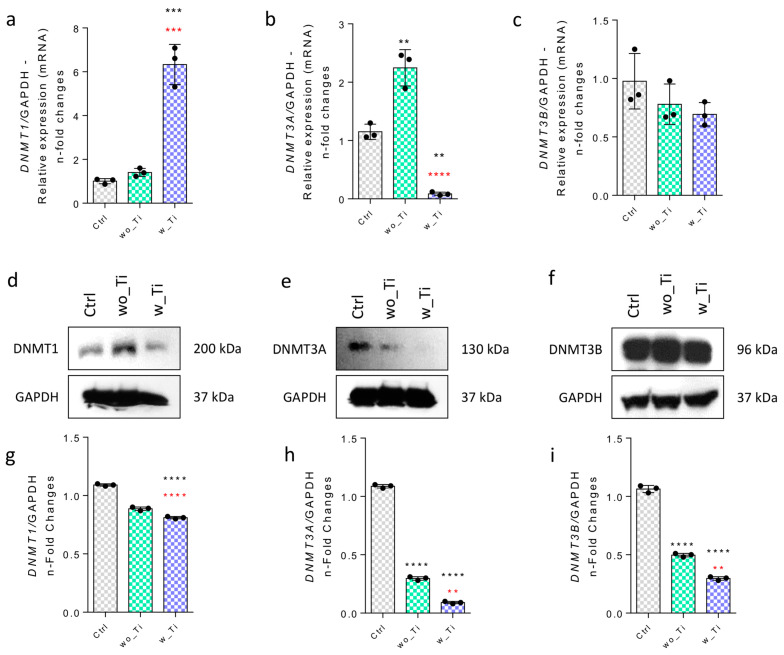
Effect of titanium-enriched medium on DNMTs. Firstly, the gene activities were evaluated by qPCR, as follows: DNMT1 (**a**), DNMT3A (**b**), and DNMT3B (**c**), where GADPH gene was used to normalize the data as a housekeeping gene. Thereafter, the protein profile was evaluated by Western blot by resolving 75 μg of sample per lane and later incubating it with specific antibodies. Representative immunoblots of DNMT1 (**d**,**g**), DNMT3A (**e**,**h**), DNMT3B (**f**,**i**) are displayed. Densitometric analysis of immunoblots was normalized to the protein ratio of controls and GAPDH, here used as loading control (housekeeping control). Differences were considered statistically significant when ** *p* < 0.001, *** *p* < 0.0002, **** *p* < 0.0001 when compared with the control (Ctrl) group; and ** *p* < 0.001, *** *p* < 0.0002, **** *p* < 0.0001 when comparing wo_Ti with w_Ti. The graphs show mean ± standard deviation of three independent experiments. DNMT: DNA methyltransferase, GAPDH: glyceraldehyde-3-phosphate dehydrogenase, Ctrl: control group, wo_Ti: medium enriched with titanium without DAE, w_Ti: medium enriched with titanium with DAE. The experiment was performed in triplicate (n = 3).

**Figure 6 jfb-14-00131-f006:**
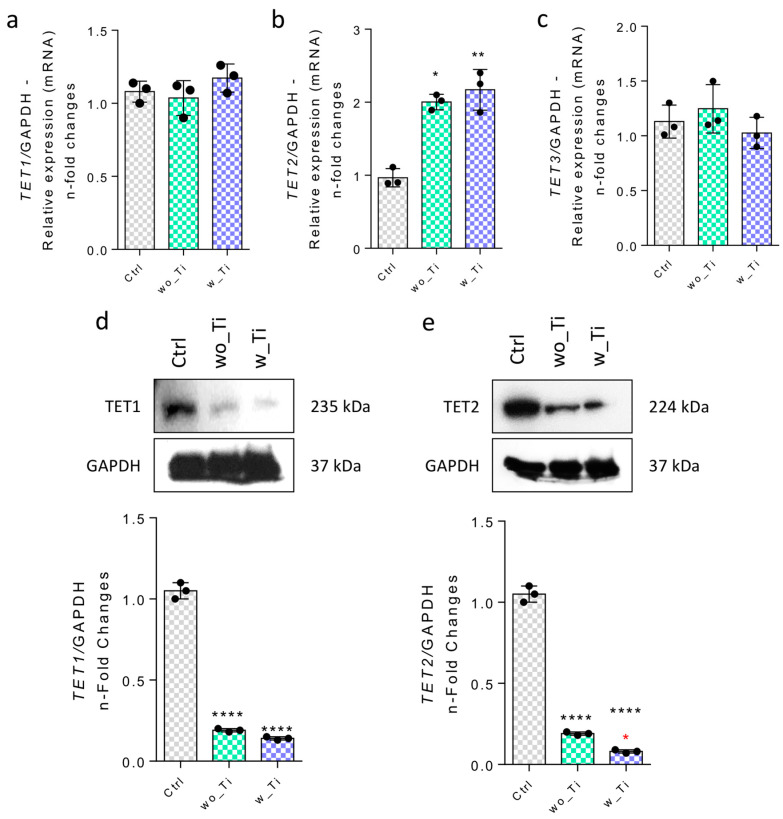
Effect of titanium-enriched medium on TETs. Firstly, the gene activities were evaluated by qPCR, as follows: TET1 (**a**), TET2 (**b**), and TET3 (**c**), where GADPH gene was used to normalize the data as a housekeeping gene. Thereafter, the protein profile was evaluated by Western blot by resolving 75 μg of sample per lane and later incubating it with specific antibodies. Representative immunoblots of TET1 (**d**) and TET2 (**e**) are presented. Densitometric analysis of immunoblots was normalized to the protein ratio of controls and GAPDH, here used as loading control (housekeeping control). Differences were considered statistically significant when * *p* < 0.04, ** *p* < 0.001, **** *p* < 0.0001 when compared with the control (Ctrl) group; and * *p* < 0.04 when comparing wo_Ti and w_Ti. The graphs show mean ± standard deviation of three independent experiments. Note: TET: ten-eleven translocation methylcytosine dioxygenase; GAPDH: glyceraldehyde-3-phosphate dehydrogenase; Ctrl: control group; wo_Ti: medium enriched with titanium without surface modification; w_Ti: medium enriched with titanium with surface modification. The experiment was performed in triplicate (n = 3).

**Table 1 jfb-14-00131-t001:** Data sheet of the specific genes evaluated in this study.

Gene (ID)	Primer	5′–3′ Sequence	Reaction Conditions	Product Size
HDAC1 (3065)	Forward	CTG GCC ATC ATC TCC TTG AT	95 °C—10 s; 58 °C—30 s; 72 °C—30 s	216 pb
	Reverse	ACC AGA GAC GTG GAA ACT GG		

HDAC2 (3066)	Forward	TTC TCA GTG CAC CCA GTC AG	95 °C—10 s; 59 °C—30 s; 72 °C—30 s	170 pb
	Reverse	CCA GTA TCC TTG GGG GAA AT		

HDAC3 (8841)	Forward	ACG TGG GCA ACT TCC ACT AC	95 °C—10 s; 58 °C—30 s; 72 °C—30 s	219 pb
	Reverse	GAC TCT TGG TGA AGC CTT GC		

HDAC6 (10013)	Forward	AAG TAG GCA GAA CCC CCA GT	95 °C—10 s; 59 °C—30 s; 72 °C—30 s	416 pb
	Reverse	GTG CTT CAG CCT CAA GGT TC		

SIRT 1 (23411)	Forward	GCA GAT TAG TAG GCG GCT TG	95 °C—15 s; 60 °C—30 s; 72 °C—30 s	152 pb
	Reverse	TCT GGC ATG TCC CAC TAT CA		

DNMT1 (1786)	Forward	AGG ACC CAG ACA GAG AAG CA	95 °C—15 s; 60 °C—30 s; 72 °C—30 s	201 pb
	Reverse	GTA CGG GAA TGC TGA GTG GT		

DNMT3A (1788)	Forward	AGG AAG CCC ATC CGG GTG CTA	95 °C—15 s; 60 °C—30 s; 72 °C—30 s	225 pb
	Reverse	AGC GGT CCA CTT GGA TGC CC		

DNMT3B (1789)	Forward	TCG ACT TGG TGG TTA TTG TCT G	95 °C—15 s; 60 °C—30 s; 72 °C—30 s	129 pb
	Reverse	TCG AGC TAC AAG ACT GCT TGG		

TET1 (80312)	Forward	GCC CCT CTT CAT TAC CAA GTC	95 °C—15 s; 60 °C—30 s; 72 °C—30 s	211 pb
	Reverse	CGC CAG TTG CTT ATC AAA ATC		

TET2 (54790)	Forward	GGT GCC TCT GGA GTG ACT GT	95 °C—15 s; 60 °C—30 s; 72 °C—30 s	245 pb
	Reverse	GGA AAA TGC AAG CCC TAT GA		

TET3 (200424)	Forward	GGT CAG GCT GGT TTA CAA CG	95 °C—15 s; 60 °C—30 s; 72 °C—30 s	198 pb
	Reverse	GGC ATA GAC CCA CAC ACA TCT		

GAPDH (2597)	Forward	AAG GTG AAG GTC GGA GTC AA	95 °C—10 s; 58 °C—30 s; 72 °C—30 s	345 pb
	Reverse	AAT GAA GGG GTC ATT GAT GG		

## Data Availability

Not applicable.

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
