# Peer review of "Titanium-Enriched Medium Promotes Environment-Induced Epigenetic Machinery Changes in Human Endothelial Cells"

_jfb, 2023, doi:10.3390/jfb14030131_

Round 1
Reviewer 1 Report
General comment:
The authors reported the effect of the culture with titanium-enriched medium on endothelial cells, which is very interesting and important topic for the development of biomaterials. The authors may think that the value of this study is mainly for implantable biomaterials, and I also agree with that. However, studying the effect of titanium-enriched medium on endothelial cells might have more impact than the way the authors think. I value this study and believe the knowledge the authors got should be published. However, for acceptance, the authors should modify the manuscript, and perhaps the experiments should be added. Especially, the trial number was not shown, which is a problem. If the trial number is only 1, the authors need to perform more experiments.
Major comment 1:
About Figs. 2 and 3. I cannot find the information of the trial numbers. If the trial number is only 1, the authors need to perform more experiments. If the author have several experimental trials, you can show the result with dotplot.
Major comment 2:
I think your result may be very interesting. However, I cannot perfectly understand the experimental design. What do control, TiO2/Wo, and TiO2/Wo mean? I’m not sure what does “treatment” stand for. You can show the experimental conditions by using figure or Table including culture time.
Major comment 3:
The author may think these results can be used for the development of implantable biomaterials. I believe the results you reported is useful for bioreactors also. Nowadays, many bioreactors or culture vessels have been reported [DOI: 10.1089/ten.tec.2011.0717; DOI: 10.1093/jbcr/irab034], and some of them use titanium or other metallic materials [DOI: 10.1116/1.4907754; DOI: 10.1038/s41598-021-00908-0; DOI: 10.1016/j.jwpe.2019.101093; DOI: 10.1007/s12257-022-0227-1]. The effect of the use of titanium is very important for the bioengineering research flow. You should touch these in the discussion part or introduction part.
Major comment 4:
Line 260: “Previously, we have shown that titanium devices induce a dynamic microenvironment changes by releasing titanium molecules, which are very active in bone cells.”
You need to cite the paper.
Major comment 5:
Line 295: “As we have shown previously that titanium-enriched medium acts on cytoskeleton rearrangement, these results corroborate with our early data using osteoblasts.” You need to cite the paper.
Major comment 6:
How does this study affect the researcher for the surface modification of implantable biomaterials such as [DOI: 10.1016/j.artd.2021.12.003]? Can you discuss it?
Major comment 7:
Do the authors think this result can be used for the field of DDS?
Major comment 8:
In conclusion, the authors mention “Finally, the vasculature preceding the bone formation guarantees also the migration of mesenchymal stem cells to the reactional tissue which are responsible to deposit a new bone.” I do not get this. Could you explain?
Major comment 9:
In conclusion, the authors also mention “In an opposite scenario, the implanted devices will be surrounded by a fibrous capsule resulting in the failure of the implants.”. I could not follow this. Can you explain?
Minor comment 1:
Line 128: the font looks different from other parts. You can modify it.
Minor comment 2:
Line 137: the font looks different from other parts. You can modify it.
Minor comment 3:
Line 140: the font looks different from other parts. You can modify it.
Minor comment 4:
Line 182: “Thus, the Fig 2 brings the panorama analysis of mechanisms related with DNA methylation and histone acetylation processes by exemplifying genes encoding main enzymes 183 involved with epigenetic mechanism.” You do not have to put “the” before Fig. 2.
Minor comment 5:
About Fig. 4, you can show the all images of WB (uncropped images and the other images, not only typical one.) as Supplemental information.
Author Response
Feb 14, 2023.
Dear Editor:
Thank you for the attention you have given to our manuscript originally called “Titanium-enriched medium promotes environmental-induced epigenetic changes in human endothelial cells” (jfb-2195344). Indeed, we appreciated the concerns and criticism of the reviewers and the opportunity you have given us to submit a revision of our reworked manuscript.
Importantly, we addressed all the points raised in full, and feel that by incorporating their suggestions our manuscript has been much improved.
Our responses to the raised issues are summarized as follows:
REVIEWER 1:
The authors reported the effect of the culture with titanium-enriched medium on endothelial cells, which is very interesting and important topic for the development of biomaterials. The authors may think that the value of this study is mainly for implantable biomaterials, and I also agree with that. However, studying the effect of titanium-enriched medium on endothelial cells might have more impact than the way the authors think. I value this study and believe the knowledge the authors got should be published. However, for acceptance, the authors should modify the manuscript, and perhaps the experiments should be added. Especially, the trial number was not shown, which is a problem. If the trial number is only 1, the authors need to perform more experiments.
Reaction. Dear reviewer, many thanks for the unbiased evaluation of our study. We have now implemented all of these concerns properly. This new version of this study brings in detail all the points raised, in special the number of replicates carried out in our experimental design (n=3).
Major comment 1:
About Figs. 2 and 3. I cannot find the information of the trial numbers. If the trial number is only 1, the authors need to perform more experiments. If the author have several experimental trials, you can show the result with dotplot.
Reaction. Many thanks for bringing this up. All the experiments performed in this study were carried in triplicate. We have also reorganized the graphs by considering the dotplot format and in fact it becomes clearer.
Major comment 2:
I think your result may be very interesting. However, I cannot perfectly understand the experimental design. What do control, TiO2/Wo, and TiO2/Wo mean? I’m not sure what does “treatment” stand for. You can show the experimental conditions by using figure or Table including culture time.
Reaction. In this reworked manuscript this information was properly addressed. To note, we have tested here 2 conditions of titanium based on their surface’s modifications [with dual-acid etching (DAE) or not]. We have previously shown that there is difference on the releasing of titanium molecules considering this both conditions. A more appropriate description is now inserted within the manuscript.
Major comment 3:
The author may think these results can be used for the development of implantable biomaterials. I believe the results you reported is useful for bioreactors also. Nowadays, many bioreactors or culture vessels have been reported [DOI: 10.1089/ten.tec.2011.0717; DOI: 10.1093/jbcr/irab034], and some of them use titanium or other metallic materials [DOI: 10.1116/1.4907754; DOI: 10.1038/s41598-021-00908-0; DOI: 10.1016/j.jwpe.2019.101093; DOI: 10.1007/s12257-022-0227-1]. The effect of the use of titanium is very important for the bioengineering research flow. You should touch these in the discussion part or introduction part.
Reaction. We have now articulated better these suggestions along the discussion section of this reworked manuscript and the manuscript has been much improved.
Major comment 4:
Line 260: “Previously, we have shown that titanium devices induce a dynamic microenvironment changes by releasing titanium molecules, which are very active in bone cells.”
Reaction. A paper is now cited as requested.
Major comment 5:
Line 295: “As we have shown previously that titanium-enriched medium acts on cytoskeleton rearrangement, these results corroborate with our early data using osteoblasts.” You need to cite the paper.
Reaction. A paper is now cited as requested.
Major comment 6:
How does this study affect the researcher for the surface modification of implantable biomaterials such as [DOI: 10.1016/j.artd.2021.12.003]? Can you discuss it?
Reaction. We have now articulated this paper along the introduction section. Many thanks!
Major comment 8:
In conclusion, the authors mention “Finally, the vasculature preceding the bone formation guarantees also the migration of mesenchymal stem cells to the reactional tissue which are responsible to deposit a new bone.” I do not get this. Could you explain?
Reaction. This is well-taken by the reviewer!! We have intended to explain the migration of cells to surrounding tissues which might use the blood vessels to get the regenerative sites. However, it is better articulated now in this reworked manuscript.
Major comment 9:
In conclusion, the authors also mention “In an opposite scenario, the implanted devices will be surrounded by a fibrous capsule resulting in the failure of the implants.”. I could not follow this. Can you explain?
Reaction: A better articulated conclusion is now presented.
Minor comment 1:
Line 128: the font looks different from other parts. You can modify it.
Minor comment 2:
Line 137: the font looks different from other parts. You can modify it.
Minor comment 3:
Line 140: the font looks different from other parts. You can modify it.
Minor comment 4:
Line 182: “Thus, the Fig 2 brings the panorama analysis of mechanisms related with DNA methylation and histone acetylation processes by exemplifying genes encoding main enzymes 183 involved with epigenetic mechanism.” You do not have to put “the” before Fig. 2.
Reaction. All the minor points were implemented as it was requested. Thanks!!
In conclusion, we feel that we have taken into account the concerns of the reviewer in full and that the present version of the manuscript should be acceptable for publication in Journal of Functional Materials.
Looking forward to your consideration,
WILLIAN FERNANDO ZAMBUZZI
On behalf the authors

Reviewer 2 Report
Dear Authors,
In the present manuscript the authors investigate about the influence of a Ti enriched
environment on the evolution of some important progress in the human body, at cells level. More in particular, their study is focalized on the effect of such media on the endothelial cells.
The manuscript is well organized, the experiments are well described and reasonably support the author’s final conclusion.
However, I have some concerns about the “toxicity”, reported by the author in the manuscript, since Ti based materials inside the human body are usually employed for different purpose, as implantable devices, etc. How the authors explain this fact, if such type of materials are dangerous for the body? Which direction has to be take in this case?
Some language errors have to be corrected along the manuscript.
Author Response
Feb 14, 2023.
Dear Editor:
Thank you for the attention you have given to our manuscript originally called “Titanium-enriched medium promotes environmental-induced epigenetic changes in human endothelial cells” (jfb-2195344). Indeed, we appreciated the concerns and criticism of the reviewers and the opportunity you have given us to submit a revision of our reworked manuscript.
Importantly, we addressed all the points raised in full, and feel that by incorporating their suggestions our manuscript has been much improved.
Our responses to the raised issues are summarized as follows:
REVIEWER 2:
In the present manuscript the authors investigate about the influence of a Ti enriched environment on the evolution of some important progress in the human body, at cells level. More in particular, their study is focalized on the effect of such media on the endothelial cells.
The manuscript is well organized, the experiments are well described and reasonably support the author’s final conclusion.
Reaction. Many thanks for evaluating our study. It’s a pleasure considering your criticism in this reworked manuscript.
However, I have some concerns about the “toxicity”, reported by the author in the manuscript, since Ti based materials inside the human body are usually employed for different purpose, as implantable devices, etc. How the authors explain this fact, if such type of materials are dangerous for the body? Which direction has to be take in this case?
Reaction. We are in agreement with this reviewer and sorry if some misinterpretation was generated from our original version of the manuscript as it was. Now, it is better articulated in order to reorganize these statements along the manuscript. Hope you understand our comment and accept our reworked manuscripts.
Some language errors have to be corrected along the manuscript.
Reaction. Dear reviewer, sorry for this inconvenience. We have now corrected the English language as it was requested and the manuscript is much better.
In conclusion, we feel that we have taken into account the concerns of the reviewer in full and that the present version of the manuscript should be acceptable for publication in Journal of Functional Materials.
Looking forward to your consideration,
WILLIAN FERNANDO ZAMBUZZI
On behalf the authors

Round 2
Reviewer 1 Report
Basically, authors answered my questions. However, I am still confused for the meaning of “treatment” in major comment 2. Please add the explanation in your manuscript before the publication. Other than that, I feel the manuscript got improved.
Author Response
Feb 16, 2023.
Dear Editor:
Thank you for the attention you have given to our manuscript originally called “Titanium-enriched medium promotes environmental-induced epigenetic changes in human endothelial cells” (jfb-2195344). Indeed, we appreciated the concerns and criticism of the reviewers and the opportunity you have given us to submit a revision of our reworked manuscript.
Importantly, we addressed all the remaining point raised in full, and feel that by incorporating the suggestion our manuscript has been much improved in reading and comprehension.
Our responses to the raised issues are summarized as follows:
REVIEWER 1:
Basically, authors answered my questions. However, I am still confused for the meaning of “treatment” in major comment 2. Please add the explanation in your manuscript before the publication. Other than that, I feel the manuscript got improved.
Reaction. Many thanks for analyzing our study with criticism. Sorry for having not addressed this concern properly. When the “treatment” was mentioned along the text it refers to the subjection of the titanium to DAE (dual acid-etching). This is widely applied in dental implants to promote roughness in the surface. In this way, we have changed the word “treatment” along the text to “DAE” in order to better attend this request, which we fully are I agreement with this reviewer. We have better described the experimental design by solving this concern along the text in this reworked manuscript and hope this version is acceptable in this second round.
In conclusion, we feel that we have taken into account the concerns of the reviewer in full and that the present version of the manuscript should be acceptable for publication in Journal of Functional Materials.
Looking forward to your consideration,
On behalf of all authors,
WILLIAN FERNANDO ZAMBUZZI
On behalf the authors
